# A Study on Magnetic Removal of Hexavalent Chromium from Aqueous Solutions Using Magnetite/Zeolite-X Composite Particles as Adsorbing Material

**DOI:** 10.3390/ijms21082707

**Published:** 2020-04-14

**Authors:** Maria-Elisavet Kouli, George Banis, Maria G. Savvidou, Angelo Ferraro, Evangelos Hristoforou

**Affiliations:** 1Laboratory of Electronic Sensors, School of Electrical and Computer Engineering, NTU of Athens, Iroon Polytechniou 9, 15780 Zografou, Greece; gbanis@mail.ntua.gr (G.B.); ferraro@eie.gr (A.F.); hristoforou@ece.ntua.gr (E.H.); 2Biotechnology Laboratory, School of Chemical Engineering, NTU of Athens, 9 Iroon Polytechniou Str., Zografou Campus, 15780 Athens, Greece; savvid.maria3@gmail.com

**Keywords:** magnetic removal, hexavalent chromium, magnetite, zeolite-X, composite nanoparticles, adsorption

## Abstract

Toxic and heavy metals are considered harmful derivatives of industrial activities; they are not biodegradable and their accumulation in living organisms can become lethal. Among other heavy and toxic metals, chromium is considered hazardous, especially in the hexavalent (Cr^6+^) form. Numerous established studies show that exposure to Cr^6+^ via drinking water leads to elevated chromium levels in tissues, which may result in various forms of cancer. The purpose of this research is to synthesize magnetite/zeolite-X composite particles for the adsorption and magnetic removal of Cr^6+^ ions from aqueous solutions. Synthesis and characterization of such composite nanomaterials, along with an initial experimental evaluation of Cr^6+^ removal from water-based solution, are presented. Results show that zeolite-X is a very promising zeolite form, that when bound to magnetic nanoparticles can be used to trap and magnetically remove toxic ions from aqueous solutions.

## 1. Introduction

The use of nanomaterials in biomedicine and environmental technology has become a state-of-the-art object of research recently. Concerning environmental pollutant remediation and in order to reduce biohazardous effects, several technologies, including chemical precipitation [1,2], ion exchange [3,4,5,6], reverse osmosis [7,8,9] and adsorption [10,11,12,13,14,15], have been used to remove heavy metal ions from various aqueous solutions [16]. Among these methods, adsorption has increasingly received much attention in recent years due to its unique benefits, especially when wastewater with low metal ion concentration must be treated. Many different kinds of adsorbents have been developed and among them, activated carbon [17,18,19,20,21], activated alumina [22,23,24,25], coated silica gel [26,27] and treated sawdust [28,29,30] are the most used. The objective of the present research is to implement an inexpensive and highly adsorbing mechanism based on coupling two different materials, zeolite and magnetite, into the same particle forming a composite tool that can be used for magnetic removal of harmful ions.

In this work, the adsorption properties of magnetite/zeolite-X composite particles on dissolved Cr^6+^ ions in aqueous solution are evaluated and the influence of the zeolite’s ion exchange ability during the process is investigated by performing a series of ion analysis measurements. The deviations in ion concentration are indicative of deionization and of ion exchange ability of zeolite. Ion analysis was conducted on samples containing an aqueous solution of potassium dichromate and zeolite-X powder, under constant stirring. The adsorbing material investigated is zeolite-X, a very commonly found zeolite which was bound to magnetite nanoparticles via microwave-assisted co-precipitation [31,32,33,34]. Zeolites are aluminosilicate minerals with various pore sizes whose chemical composition can be generally described by the following empirical molecular formula:
M_2z/n_ O_z_ Al_2_O_3_∙ySiO_2_∙wH_2_O(1)
where y ≥ 2/n is the valence of the equilibrating cations M, and w are the water molecules contained in the zeolite crystalline matrix. A simple criterion for distinguishing zeolites and zeolite-like materials from the densest silicate materials is the FD, the number of tetrahedral coordinated frame atoms (T atoms) per 1000 Å^3^. A gap is clearly identifiable between zeolite-type materials and dense tetrahedral frame structures. The maximum FD value for zeolite ranges from 19 to over 21 T atoms per 1000 Å^3^, depending on the type of smaller ring available, while the minimum for dense structures ranges from 20 to 22 atoms [35]. In the crystalline lattice of zeolites, each aluminum atom generates an excess of negative charge which is equilibrated by several interchangeable cations, such as H^+^, Na^+^, K^+^, Ca^2+^ and Mg^2+^. These free cations as well as water molecules are held loosely and can be moved and exchanged with other cations of the environment without substantial alteration of the crystal structure [36]. Thus, Brønsted acidic domains (proton donors) are created and scattered within the frame. Zeolites exhibit selective adsorption ability due to their molecular dimensions and channels [37] and large special surface, thanks to a microporous structure, as well as structural and thermal ability up to 1000 °C.

Zeolites have two main properties that make them ideal candidates in ion-removing applications: adsorption and ion exchange. These two properties are due to the presence of Al^3+^ with Si^4+^ ion which resides on adsorption sites, and the micropores crystalline system, which allows ion exchange. It is important to mention that zeolites are essentially nontoxic and pose no environmental risk [38].

Ion exchange depends on solid and aqueous phase composition and is a function of solution concentration. Ion exchange equilibria occur between two or more phases, one of which is usually liquid and exchanges two or more ions (cations or anions) relatively strongly bound to the other phases. An ion’s exchangeable quantity by a solid exchanger depends on its structural features and is called the ion exchange capacity, usually expressed in mEq/g. Ion transfer from one phase to the other is subject to the observance of electroneutrality and is regulated by the ion concentration in both phases. This parameter is a function of both the energy of ion lattice interaction and the hydration energy (ion solution interaction).

A cation exchange reaction may be written as:
nM (m_s_) + − } − mN” (z+) + − −nM (m_z_) + + mN” (s+)(2)
where m and n are the valences of exchanging cations. M and N and subscripts s and z denote solution and zeolite phase, respectively.

Ion exchange property of zeolites results from the presence of extra cations located on channels and cages of it [39]. The zeolite structures present various cation sites, which differ from each other in the framework position and therefore in bond energy (Figure 1). When zeolite crystals are in contact with an electrolytic solution, cations escape from their sites and are replaced by other cations from the solution [40].

Zeolite-X is a synthetic zeolite which belongs to the family of aluminosilicate molecular sieves with a faujasite–frame type structure (FAU). It is called a molecular sieve due to its unique adsorbing ability. Zeolite-X is described by the following chemical formula [41]:
**|**(Ca^2+^,Mg^2+^Na^+^_2_)_29_ (H_2_O)_240_**|[**Al_58_Si_134_O_384_**]** − FAU (3)

It is important to note that the lower the electronegativity (strong positive) of the metal cation (or the closer its atomic diameter to the pore diameter), the stronger the bond it forms with the zeolite Thus, the one-way ion exchange of heavy cations with alkali and alkaline earths into zeolite pores is favored. In Figure 2, the framework structures of zeolite-X and faujasite-like zeolites are presented.

Magnetite, which constitutes the magnetic material of the proposed composite material, is commonly known as iron oxide, a strong ferromagnetic material with the chemical formula Fe_3_O_4_. When it is synthesized in nanoparticles < 50 nm it exhibits superparamagnetic behavior at room temperature. In magnetite, the O^2−^, Fe^2+^ and Fe^3+^ ions form an inverse spinel crystallographic structure [44]. The face-centered cubic spinel structure of magnetite is illustrated in Figure 3.

The composite magnetic particles that were synthesized in this study resemble a sponge-like material composed of a zeolitic matrix with surficial adsorbed magnetite nanoparticles. During the synthesis procedure the nanosized magnetic particles are adsorbed by the surface of the zeolite-X and enclosed inside the pores and channels of the matrix material, providing the final composite form.

## 2. Results and Discussion

The adsorption properties of magnetite/zeolite-X composite particles on dissolved Cr^6+^ ions in aqueous solution are evaluated and the influence of the zeolite’s ion exchange ability during the process is investigated by performing ionic analysis measurements. Ion analysis was conducted on samples containing an aqueous solution of potassium dichromate and zeolite-X powder, under constant stirring. The adsorbing material investigated is aeolite-X, a common form of zeolite which was bound to magnetite nanoparticles via microwave-assisted co-precipitation [36,37,38,39].

### 2.1. Structural Analysis

#### 2.1.1. Scanning Electron Microscopy (SEM)

The final product was studied macroscopically (Figure 4e) and microscopically by scanning electron microscopy (SEM) as illustrated in Figure 4a–c. The morphology of the pure zeolite-X particles prepared is illustrated in Figure 4d. The morphology consists of uniform spheroidal cubic particles with average diameter less than 10 μm. The resulting color powder was grayish-black (Figure 4e), as opposed to pure magnetite whose color is bright black, indicating the complexation of magnetite with zeolite-X to form the composite particles.

The size of the magnetite nanoparticles was in the range of 40–70 nm, small enough to enter and form complexes with zeolite-X and give a final size for the composite particles of around 10 μm with a morphology of granules, mainly spheroidal with some angular expressions. It should be noticed that the particle size range, especially for the case of the composite particles, occurs as a result of unavoidable agglomeration due to air humidity and prolonged storage time, given that zeolites can absorb humidity. The shape of the composite particles differs from the pure zeolite as it is the result of surficial adsorption.

#### 2.1.2. X-ray Diffraction (XRD)

The crystalline structure of the particles produced was analyzed by X-ray diffraction (XRD). XRD patterns were obtained with a BRUKER-binary V4 (RAW) diffractometer using CrKα radiation (the anode material was Cr). The crystallite sizes were estimated from the line broadening of the (422)/54°—for the magnetite—and at the 45°—for the composite nanoparticle. The peaks were obtained using a scanning step size of 0.0250° and scan step time of 368 sec. The crystallite size D was calculated from the Scherrer—Debye equation,
(4)$$D=\frac{K\lambda }{\beta *cos\Theta }$$
where *K* = 0.9, *λ* is the wavelength of the X-rays, *Θ* is the diffraction angle (*Θ* = 2θ/2) and *β* is the corrected full width at half-maximum (FWHM) of the peak given by
(5)$$\beta =\beta_{m}^{2}-\beta_{s}^{2}$$
where *K* is a dimensionless shape factor, with a value close to unity. The shape factor has a typical value of about 0.9, 2.29 Å for Cr K-Alpha1 radiation, *β_m_* is the measured half-width, and *β_s_* is the half-width of a standard magnetite/zeolite-X sample with a crystalline size larger than 1 μm (which is illustrated in an XRD pattern as a straight line instead of a peak). Therefore, for magnetite the Equation (1) takes the form
(6)$$D=\frac{0.9*2.29Å}{0.0227rad*cos27}\approx 10\;\mathrm{nm}$$


And for the magnetite in the zeolite-X composite nanoparticle, Equation (1) transforms to:
(7)$$D=\frac{0.9*2.29Å}{0.017rad*cos22,5}\approx 13\;\mathrm{nm}$$


And, relevantly, for zeolite-X the crystallite size was calculated as 800 nm.

The XRD spectrum for the composite magnetite/zeolite-X particle is illustrated in Figure 4b while Figure 5c illustrates the pure zeolite-X’s XRD spectrum. The results of XRD measurements are obtained when the material in the sample receives X-rays of very high energy density at specific angles (2θ) and for a certain range of angles between the transmitter and receiver of the machine. X-rays induce vibration of specific atomic bonds of the material, which diffract the X-rays towards the receiver. The specific deflection angles can be used to identify materials since they are unique for each atomic bond.

Figure 5a illustrates XRD results for the synthesized magnetite. The peaks at 18°, 27° (slightly displaced to the left), 40°, 45°, 54°, 57°, 66° (slightly displaced to the right), 73°, 84° and 91°, indicating the cubic structure of the inverse spinel, are clearly evident in this diagram, which confirms the successful production of magnetite by the microwave-assisted co-precipitation method.

In the spectrum of Figure 5b, the 15°, 20°, 22°, 24°, 26°, 30°, 34°, 36°, 40° and 46° peaks indicate the presence of zeolite-X. The peaks at 27°, 40°, 45°, 54°, 57° and 66°, indicating the cubic structure of the inverse spinel, are also evident in this diagram, which confirms the successful binding of magnetite nanoparticles to zeolite-X by microwave-assisted co-precipitation method. For Figure 5c, the peaks at 7°, 10°, 12.5°, 16°, 20.5°, 22°, 24°, 26°, 30° and 34° are typical for the specific crystalline sodium silicate material. The composite particles prepared were found to consist of 68% magnetite and 32% zeolite X.

#### 2.1.3. Fourier Transformation Infrared (FTIR)

Further evidence of the successful preparation of composite materials was provided by FTIR spectra. In Figure 6, the FTIR spectra for the magnetite and composite zeolite-X are illustrated. The FTIR evaluation provides information based on the compounds’ chemistry. It is a very helpful XRD auxiliary method for assessing inorganic substances and the eligible method used to identify organic compounds.

Referring to the magnetite phase in the composite particle spectra, the absorption band in the 580 cm^−1^ region corresponds to the characteristic vibration of the Fe-O-Fe bond of the magnetite nanoparticles (organometallic bond). We note that the typical absorbance at 580 cm^−1^ appears in all samples (since they contain magnetite). The peaks shown at 2900 cm^−1^ and 2850 cm^−1^ are due to the vibrations of the O-H bonds of the absorbed water molecule as well as to the vibrations of the O-H bonds attached to the surface of the magnetite nanoparticles because of their way of preparation through ammonia hydrate. The peak at 1637 cm^−1^ is attributed to the vibrations of the O-H bonds of the absorbed water molecules. The peaks at 1400 cm^−1^ are attributed to the filler (KBr pellet).

Referring to the zeolite-X spectrum of composite particles, the characteristic peaks at 3400 cm^−1^ (OH group) and 1637 cm^−1^ are due to the water molecules associated with Na and Ca in the channels of zeolite’s pores. Additional spectra appear near 1067 cm^−1^ and 796 cm^−1^. The band of 1067 cm^−1^ corresponds to the asymmetric states of extensional oscillations of the internal T-C bonds in the tetrahedrons TO_4_ (T = Si and Al). The oscillations at 796 cm^−1^ are attributed to the oscillation states of the O-T-O groups. The FTIR spectra obtained come in agreement with the XRD spectra confirming the successful binding of the zeolite and the magnetic nanoparticles in the composite material.

#### 2.1.4. Vibrating Sample Magnetization 

Mass magnetization saturation was determined by vibrating sample magnetometry-VSM at room temperature, since temperature is a factor that highly affects the hysteresis loop of magnetic nanoparticles, metals and metallic alloys [46,47,48,49]. Two representative magnetization curves of nanoparticles prepared with and without zeolite-X are illustrated in Figure 7.

Through the magnetic hysteresis loop characterization, it is observed that both samples show similar magnetic behavior. Magnetite without zeolite-X exhibits a 70 Am^2^ kg^−1^ saturated mass magnetization which is reached in lower magnetic fields < 1.5 T, while magnetite with zeolite-X exhibits a 29 Am^2^ kg^−1^ saturated mass magnetization which is reached in higher magnetic fields > 1.5 T. The lower magnetic moment of the composite particles is an outcome of the complexation of pure magnetite with zeolite-X, as the latter reduces the magnetic properties of the magnetite. The magnetization loops obtained at room temperature exhibit an absence of hysteresis and residual magnetization within the limits of our experimental uncertainty, which is the main characteristic of superparamagnetic behavior of a few nm particle size [50,51,52,53]. It is worth mentioning that the superparamagnetic behavior is the ability of ferromagnetic and ferrimagnetic materials with sizes smaller than 50 nm to get magnetized when an external magnetic field is applied and loses any residual magnetization (instant demagnetization) when the field is removed. In magnetic removal and magnetic manipulation applications, it is essential to use nanoparticles that exhibit this behavior as it is easy to manipulate them in any direction by applying a standard magnetic field. It is very important though that the size of the nanoparticles does not cross a lower limit of 10nm. Under this size limit the nanoparticles exhibit great difficulties to be controlled magnetically. Furthermore, the superparamagnetic behavior avoids particle agglomeration since in the absence of magnetic fields a solution containing magnetite nanoparticles presents colloidal characteristics.

### 2.2. Ion Analysis

The main objective of the present study was the implementation of an integrated method for the magnetic removal of hexavalent chromium from aqueous systems. As presented, the first part included the chemical characterization of composite particles while the second part investigates the adsorbing ability of zeolite-X. To do so, ion analysis was implemented in 100 mL of deionized water containing 0.2 g of potassium dichromate. Measurements were performed before and after zeolite-X treatment at different times, temperatures and zeolite concentrations. It is important to mention that the vast majority of the curves obtained exhibit a very satisfactory adsorbing behavior of the zeolite. By analyzing the following ion reduction curves, the best combination of zeolite concentration in relation to time and solution temperature is going to be investigated and evaluated.

From the charts illustrated in Figure 8 it is possible to conclude that in relation to the hexavalent chromium, ion removal from the aqueous solutions in all experimental conditions is observed at 25 °C for zeolite concentrations up to 20 g/L and that we can achieve the best adsorbing efficiency within 1 h. The time seems to be the most critical parameter, since the maximum removal at every temperature and for all different composite zeolite concentrations is achieved within the first hour of treatment and is proportional to the amount of composite zeolite-X dissolved in the solution. More specifically, Figure 8a shows chromium ion level reduction from 0.24 ppm to 0.14 ppm within the first hour and with potassium dichromate:zeolite X ratio of 2:1 g/L. Figure 8b shows the hexavalent chromium levels drop from 0.24 ppm to 0.13 ppm within the first hour with a potassium dichromate:zeolite X ratio of 2:2 g/L. In Figure 8c it can be observed that the hexavalent chromium levels decrease from 0.24 ppm to 0.11 ppm and remain stable at 0.11 ppm up to 6 h with a potassium dichromate:zeolite X ratio of 2:5 g/L. Figure 8d reports that the ion levels decrease from 0.24 ppm to 0.09 with a potassium dichromate:zeolite X ratio of 2:10 g/L. Finally, in Figure 8e it can be observed that the chromium ion levels decreased from 0.24 ppm to 0.06 ppm and then to 0.05 ppm and remained stable for up to 6 h with potassium dichromate:zeolite X ratio of 2:20 g/L. With small differences, the trend of ion removal values obtained between 5 °C and 50 °C follows a linear behavior, within 1 h the max adsorption was achieved and further reaction time did not significantly affect such trends. The experimental adsorption tests performed at 70 °C seem to exhibit a slight unpredictable fluctuation maybe due to the highly energetic conditions of the reaction. Indeed, it is possible that the thermal energy at 70 °C can cause perturbations on the ion exchange process. For higher temperatures and adsorbent concentrations, we observe instability of the ion levels during the treatment procedure. This incident stems from the enhanced cation exchangeability between the zeolite and the solution due to the increased temperature. However, temperatures of 25 °C and 5 °C represent outdoor environmental conditions during summertime and winter in a Mediterranean country and the experimental results provided seem very convenient for such a procedure in natural environmental.

All the above observations are confirmed by the hexavalent chromium reduction percentage presented in Figure 8. The percentages occurred stem from the formula:
(8)$$\%\;reduction=\frac{C_{a}-C_{b}}{C_{a}}*100$$
where *C_a_* is the initial concentration of hexavalent chromium and *C_b_* is the final concentration of hexavalent chromium in the solution after the treatment.

The chart in Figure 9a illustrates the reduction of chromium concentration in correlation to the number of treatment cycles with composite zeolite-X at T = 25 °C and pH = 5.5 for the ratios (i) 2:1 g/L, (ii) 2:2 g/L, (iii) 2:5 g/L, (iv) 2:10 g/L, and (v) 2:20 g/L of potassium dichromate:zeolite X. In all five cases, a sharp reduction in chromium ion concentration of more than 10 ppm (for 20 g/L the concentration is reduced by almost 20 ppm) is observed during the first hour of treatment. Then, for each subsequent cycle a gradual decrease of the concentration of chromium ion occurs, but is clearly lower than the first ones. For composite zeolite-X concentrations of 1 g/L and 2 g/L, the reduction of hexavalent chromium ions continues up to the 6th treatment cycle. For zeolite-X concentrations 5 g/L, 10 g/L and 20 g/L, from the 3rd treatment cycle we can observe stabilization which is kept until the 6th cycle. At this point it must be noted that after the 6th cycle the measured concentrations were 0.06–0.05 ppm less than the starting concentration (0.24 ppm). The lower ionic value was observed for the zeolite concentration of 20 g/L and the ion decrease was faster than the one observed at the four other zeolite concentrations during the first treatment cycle. Notably, the ionic concentration value dropped from 0.24 ppm to 0.06 ppm during the first cycle and reached the final value of 0.05 ppm during the third cycle and then it stabilized. The above observations are confirmed by the hexavalent chromium reduction percentage graphs shown in Figure 10a and the respective values presented in Table 3.

Figure 9b illustrates the alternation of chromium ion concentration in a time space of 1 h treatment with composite zeolite-X at T = 25 °C and pH = 5.5. It is clear that from the first ten minutes of treatment thatthe adsorbent becomes active and adsorbs a very large proportion of the chromium ions. In particular, for the smaller zeolite-X concentrations (1 g/L, 2 g/L and 5 g/L), the adsorption of hexavalent chromium ions results in ion reductions of about 0.10 ppm. For the higher adsorbent concentrations (10 g/L and 20 g/L), the magnitude of the ion depletion is much higher, reaching 0.15 ppm and, impressively for a solution with 20 g/L zeolite, the hexavalent chromium concentration reaches zero at 40 min. Then, it increases as the zeroing point is out of balance, thus the balance between the free solution cations and the zeolite bound ones (due to increased cation exchange) is restored after at least one hour of treatment. All the above observations are confirmed by the hexavalent chromium reduction percentage graphs in Figure 10b and the respective values presented in Table 4.

Table 1 shows the chromium reduction percentage in correlation to the number of treatment cycles with zeolite X at T = 25 °C and pH = 5.5 for the zeolite X concentrations: (i) 1 g/L, (ii) 2 g/L, (iii) 5 g/L, (iv) 10 g/L and (v) 20 g/L, while Table 1 illustrates the chromium reduction percentage in correlation to the time (10 min for the first treatment hour) at T = 25 °C and pH = 5.5 for the ratios: (i) 2:1 g/L, (ii) 2:2 g/L, (iii) 2:5 g/L, (iv) 2:10 g/L and (v) 2:20 g/L potassium dichromate:zeolite X.

Finally, from percentage reduction chart in Figure 10b and Table 2, it is noticeable that after 40 min of treatment with 20 g/L composite zeolite-X, the reduction rate reaches 100%. Although this percentage occurs during the unstable phase of the treatment procedure because of increased cation exchange, it is a significant result since by stopping the procedure at this crucial point the total chromium adsorption might reach a 100% efficiency.

In comparison to previous research, the results obtained by treating hexavalent chromium with faujasite-like zeolite-X proved the superior properties of the specific zeolite in heavy ion adsorbing applications. Comparing to previous research for different zeolites and their efficiency in hexavalent chromium removal, the results of the current research showed better performance in a slightly acidic environment which was naturally adjusted by the introduction of magnetite/zeolite-X particles to the chromate solution. Indeed, the highest rates of removal (70% efficiency) are reached in heavy acidic environments when FeO/natural zeolite/sucrose particles are used for 1 h [54]. In other works, it was showed that for pH = 5, the reduction rates of chromium reach 50% when using modified nano-zeolite A [55], while zeolitic membranes succeed a removal efficiency of around 45% at 3 h [56].

## 3. Materials and Methods

### 3.1. Materials

The materials used for the preparation of nanoparticles were potassium dichromate ≥ 99.8% (Honeywell, Charlotte, NC, USA), iron chloride (II) tetrahydrate 99% (Merck, Kenilworth, NJ, USA), iron chloride (III) hexahydrate 99% (RS), acetic acid 99–100% (Sigma Aldrich, St. Louis, MO, USA), sodium hydroxide 99%, alumina trihydrate (Merck, 65% Al_2_O_3_), sodium silicate solution, ammonia hydrated 30–33% (Sigma Aldrich), glycerol 98% (Chembiotin, Athens, Greece), absolute ethanol 99.99% (Fisher Scientific, Waltham, MA, USA) and deionized water.

### 3.2. Methods

#### 3.2.1. Synthesis of Zeolite-X (Linde Type X)

The synthesis of the zeolite took place according to the protocols obtained by the International Zeolite Association [57]: for the batch preparation, 100 g of sodium hydroxide are added to 100 mL of water and stirred until dissolved (Solution A). Then, 97.5 g alumina trihydrate is added to solution A and stirred at 100 °C until the alumina is fully dissolved (Solution B). Solution B is left to cool at room temperature. Then, 202.5 g of water is added to Solution B and mixed (Solution C). A total of 100 g of solution C and 612 g water and 59.12 g sodium hydroxide are mixed until the NaOH is fully dissolved (Solution D). At the same time, 219.7 g of sodium silicate solution is added to 612 g of water and 59.12 g of sodium hydroxide. The solution is stirred until everything is dissolved (Solution E). Solution D and E are combined quickly and stirred for 30 min.

For the crystallization, the final solution is placed in a polyethylene bottle with a lid and left for 8 h at 90 °C without any agitation. Then, the product is filtered and washed multiple times until the pH drops below 10. Finally, it is left overnight to dry at 100 °C. The material retrieved is a fine white powder. The post-fabrication activation takes place in an oven at 300 °C for 3 h.

#### 3.2.2. Synthesis of Fe_3_O_4_ Nanoparticles

In a beaker with 40 mL deionized water, ferric chloride (FeCl_3_·6H_2_O) and ferrous chloride (FeCl_2_·4H_2_O) are added in a 2:1 ratio and mixed. The iron chlorides are the precursors on the basis of which the reaction will take place to form Fe_3_O_4_. Afterwards, 7 mL of 0.8 M aqueous NH_4_OH is added to the solution to regulate the pH to 9. The solution enters the microwave furnace and is heated for 2.5 min at a power of 160 W. The resulting product is an aqueous two-phase solution in which the magnetite nanoparticles are present in the aqueous phase. The nanoparticles are washed 3 times with deionized water and ethanol and received via magnetic separation.

#### 3.2.3. Synthesis of Fe_3_O_4_/Zeolite-X Composite Particles

In a beaker with 10 mL deionized water, 1.5 g of zeolite-X powder, 10 mL of glycerol and 10 mL of 99.8% acetic acid are added and stirred to form a white solution. Then, the magnetite nanoparticles are added to the solution. A lid is placed on the beaker and the solution enters the microwave furnace to complete the particle synthesis process by assisted microwave co-precipitation. The heating time in microwaves is 5 min at a power of 160 W, temperature of 80 °C and with total heating energy offered 48,000 J. When the composition is completed, the nanoparticles are removed magnetically and washed five times with deionized water and ethanol successively to remove the remaining unbounded zeolite. Finally, the particles are left to dry at 70 °C, in a vacuum atmosphere, for 18–24 h.

The particles were structurally characterized by an X-ray diffractometer (2θ = 0–70° for the composite particles and 2θ = 0–90° for the magnetite nanoparticles) and visualized in particle size by SEM. Their magnetic response was studied via VSM (vibrating sample magnetometer). The determination of the percentage of each ion in water samples was performed by flame metering for untreated seawater and for treated seawater at the 1st and 6th hour of zeolite treatment for each different amount of zeolite-X and for each different temperature.

#### 3.2.4. Ion Analysis

Firstly, in order to understand the range of adsorption of the dissolved Cr^6+^ ions from the synthetic zeolite and to investigate the influence of the ion exchange capacity of the zeolite, in this process, a series of ionic analyses were performed. The instrument used for the ion analysis was the portable Smart 3 Colorimeter by La Motte. The purpose of these measurements was to evaluate the probability of deionization and the ion exchange capacity of zeolite-X. For that reason, composite nanomaterials of five different zeolite-X net weights were used: 0.1 g, 0.2 g, 0.5 g, 1 g and 2 g. The percentage of zeolite-X in the particles prepared was 40% of the total particle mass, so the nanoparticle concentration used in the solutions was calculated according to the desired net zeolite-X weight. The Cr^6+^ ion exchange measurements for solutions (a) 2:1 g/L, (b) 2:2 g/L, (c) 2:5 g/L, (d) 2:10 and (e) 2:20 g/L potassium dichromate:zeolite-X were conducted according to the protocol presented in Table 1 and Table 2. The Cr^6+^ ion exchange charts derived from the measurements are presented on Table 3. The charts resulted from a series of measurements carried out at 5 °C, 25 °C, 50 °C and 70 °C at pH = 5.5 and for a period of 1 to 6 h (Table 3). The initial pH of the solutions was 4.5 and after the treatment with zeolite increased to 5.5. After the treatment, the nanoparticles were magnetically removed from the potassium dichromate solution using a permanent neodymium magnet (NdFeB magnet grade N48). The field produced by the magnet was standardized at 0.3 T in a distance of few millimeters. Then, the solution was subjected to colorimetry testing in order to evaluate the chromium ion removal. The main advantage of this procedure is the simplicity of the particle removal from the solution (taking seconds), by applying an external magnetic field, without extra micro or nanofiltration procedures.

Secondly, the performance of applying treatment cycles to a sample is studied in order to determine the best zeolite ion adsorption results. This time, the potassium dichromate samples were treated with zeolite at all aforementioned concentrations (1, 2, 5, 10 and 20 g/L), for 1 h. Then, the hexavalent chromium ion concentration was measured and then new zeolite was added at one of the specific concentrations for each different solution sample. The procedure was repeated every hour for a total of 6 cycles. Table 4 demonstrates the experimental protocol for measurements made for the treated solutions with zeolite-X based on the influence factors.

At last, the adsorption behavior of the zeolite during the 1st treatment hour was studied. The first 60 min were of great interest as the ionic equilibrium of the solutions was not reached yet and it provided the chance to study possible larger, even instantaneous, adsorption. The hexavalent chromium ion removal charts for this case are presented in Figure 10b and Figure 11b The measurements for the 1st hour of treatment occurred for all the aforementioned zeolite-X values (1, 2, 5, 10 and 20 g/L), at 25 °C and with sampling every 10 min.

## 4. Conclusions

A multicycle water treatment involving removal of hexavalent chromium ions has been demonstrated by using a composite material containing zeolite-X, which is able to function as a very good sorbent to remove such heavy metal ions. The use of magnetic nanoparticles engaged in a zeolite-X matrix could be a very good solution for purifying heavy metal contaminated water since magnetic removal of composite particles might offer an affordable option to withdraw pollutant and adsorbent.

Microwave-assisted co-precipitation proved an effective method for the synthesis of the composite particles. The particle structure was determined via XRD, SEM and FTIR methods and their susceptibility to magnetization via VSM. The structural analysis showed high quality of composite particles. Particles were tested for hexavalent chromium ion adsorption in aqueous solutions and the results showed remarkable depletion of the hexavalent chromium.

Zeolite-X seems a very promising raw material for the purification of water contaminated by heavy metal ions. In a very narrow time frame, it is able to adsorb cations from aqueous solutions by exchanging them with the sodium cations caged in its frame type cavities, reaching purification rates near 100%.

Since in any wastewater treatment process the efficiency of adsorption and the regeneration of the adsorbent are crucially important, to further implement this methodology the composite particle regeneration and its reusability must be investigated. Furthermore, in many applications, reuse of the adsorbent through regeneration of its adsorption properties is an economic necessity. To test the regeneration and the reusability of the particles, one possibility could be the use of appropriate solvents such as NaOH, ethanol and others. Finally, any desorption/regeneration step should be conducted in a basic environment in order to protect the magnetic core from oxidation.

## Figures and Tables

**Figure 1 ijms-21-02707-f001:**
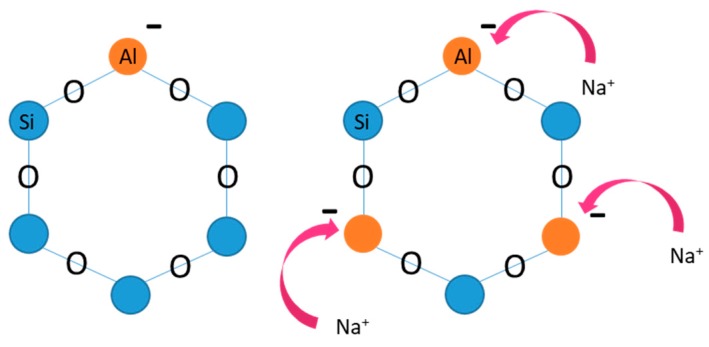
Al^3+^ and Si^4+^ cation sites. The Al-O_4_ are responsible for the excess of negative valent in Al positions and the substitution with a cation (Na, K and other).

**Figure 2 ijms-21-02707-f002:**
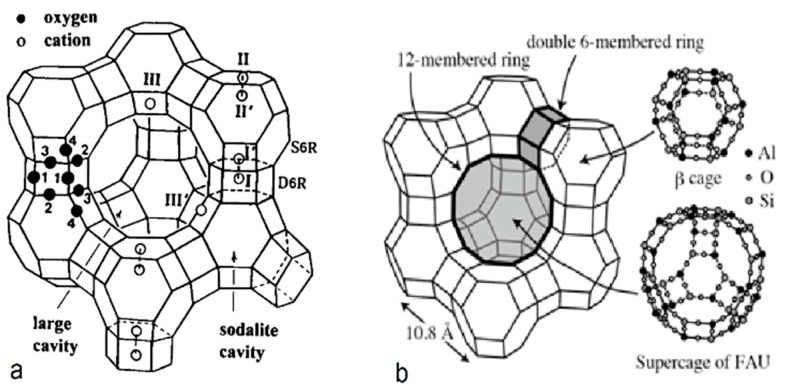
(**a**) Stylized drawing of the framework structure of zeolite-X. Near the center of each line segment is an oxygen atom. The different oxygen atoms are indicated by numbers 1–4. Silicon and aluminum atoms alternate at the tetrahedral intersections, except that Si substitutes for Al in at least 4% of the Al positions. Further long-range mixing is proposed. Extra framework cation positions are labeled with roman numerals [42]. (**b**) Schematic structure of the FAU zeolite-X framework. The small circles with a plus sign indicate possible cation sites [43].

**Figure 3 ijms-21-02707-f003:**
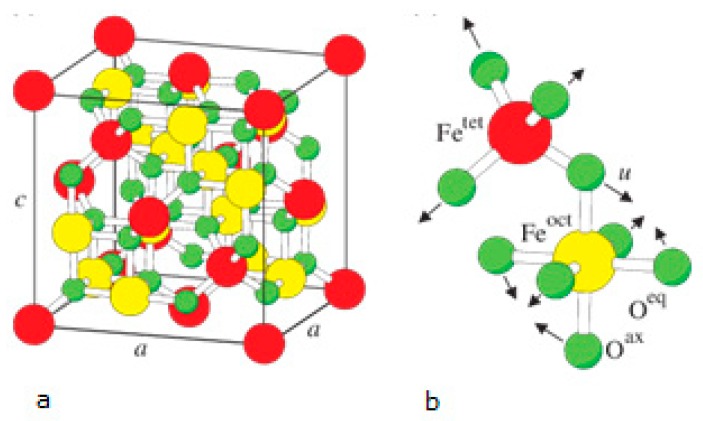
(**a**) Face-centered cubic spinel structure of magnetite. (**b**) Magnification of one tetrahedron and one adjacent octahedron sharing an oxygen atom. Large spheres labelled by Fe^tet^ and Fe^oct^ represent iron atoms tetrahedrally (tet) and octahedrally (oct) coordinated [45].

**Figure 4 ijms-21-02707-f004:**
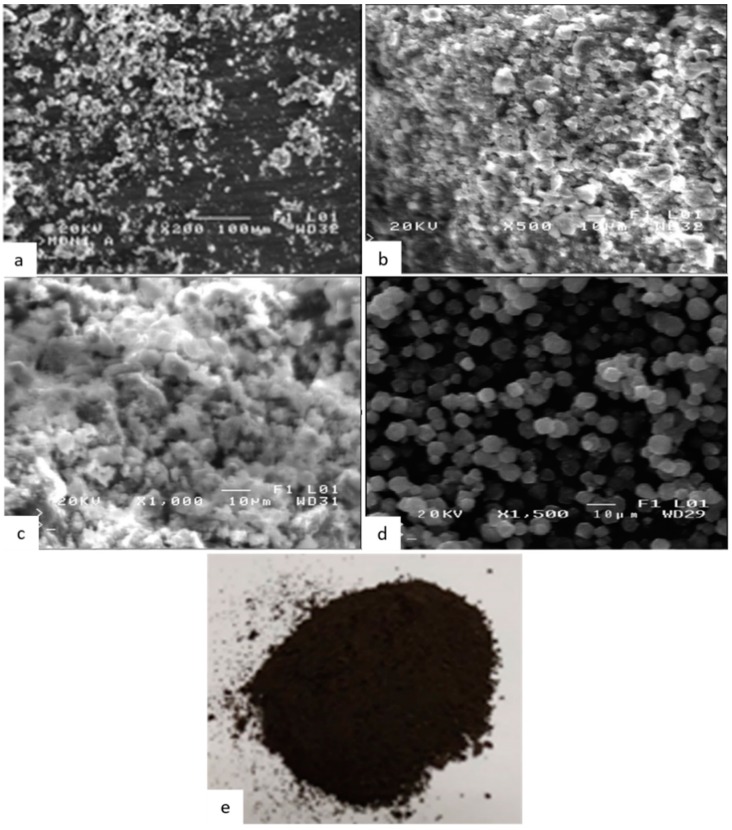
Scanning electron microscopy (SEM) analysis on the particles produced (**a**) zoom × 200, (**b**) zoom × 500, (**c**) zoom × 1000 and (**d**) pure zeolite-X synthesized and (**e**) microscopical view of the synthetized powders.

**Figure 5 ijms-21-02707-f005:**
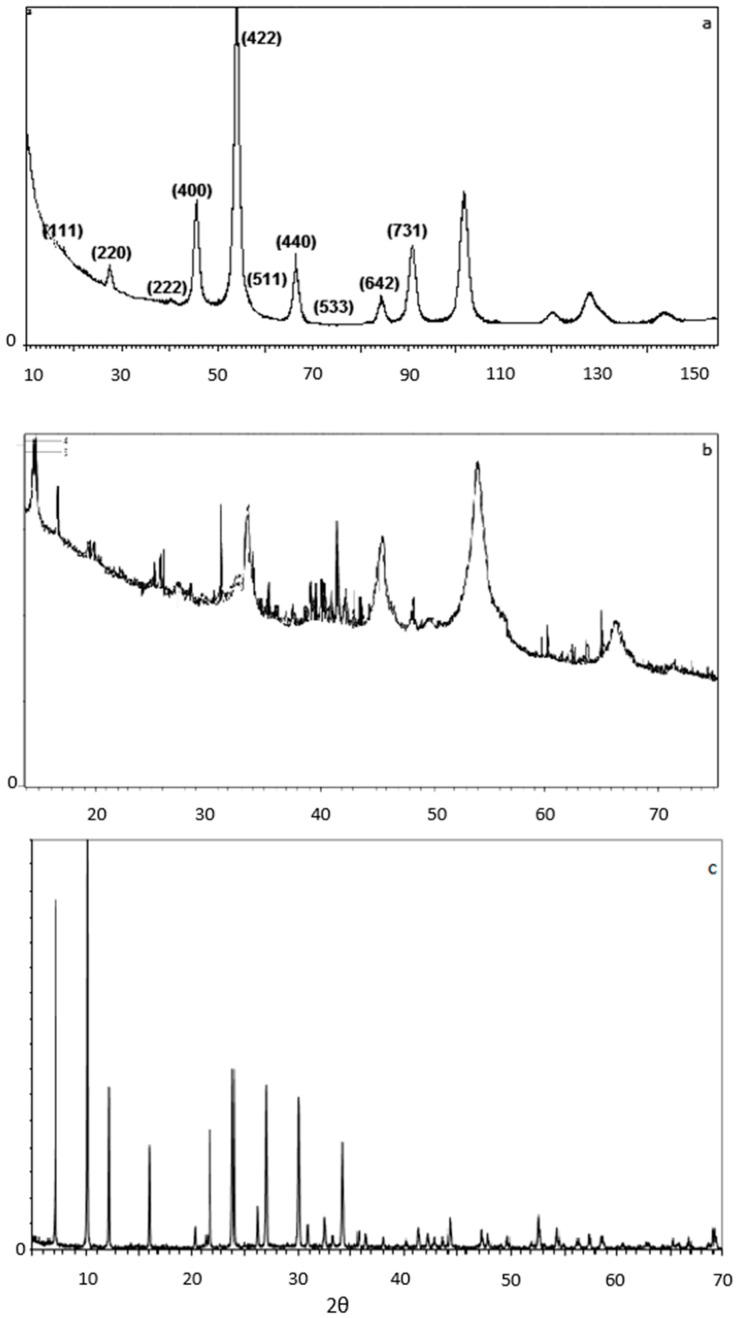
X-ray diffraction (XRD) spectra for (**a**) synthetized magnetite nanoparticles, (**b**) synthetized Fe_3_O_4_/zeolite-X composite particles and (**c**) synthesized zeolite-X.

**Figure 6 ijms-21-02707-f006:**
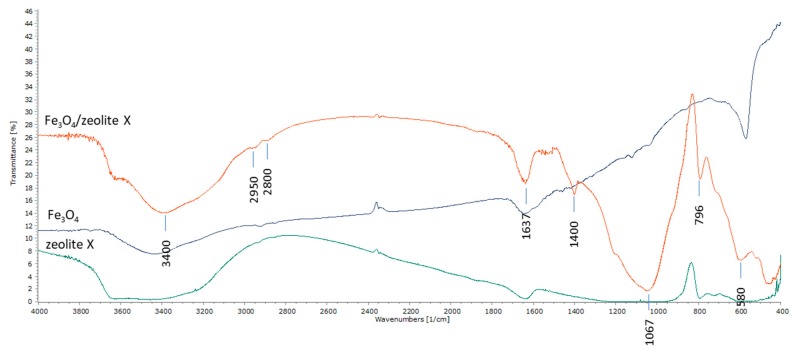
Fourier transformation infrared (FTIR) spectra for magnetite (Fe_3_O_4_)/zeolite-X composite particles, magnetite (Fe_3_O_4_) nanoparticles and pure zeolite-X sample.

**Figure 7 ijms-21-02707-f007:**
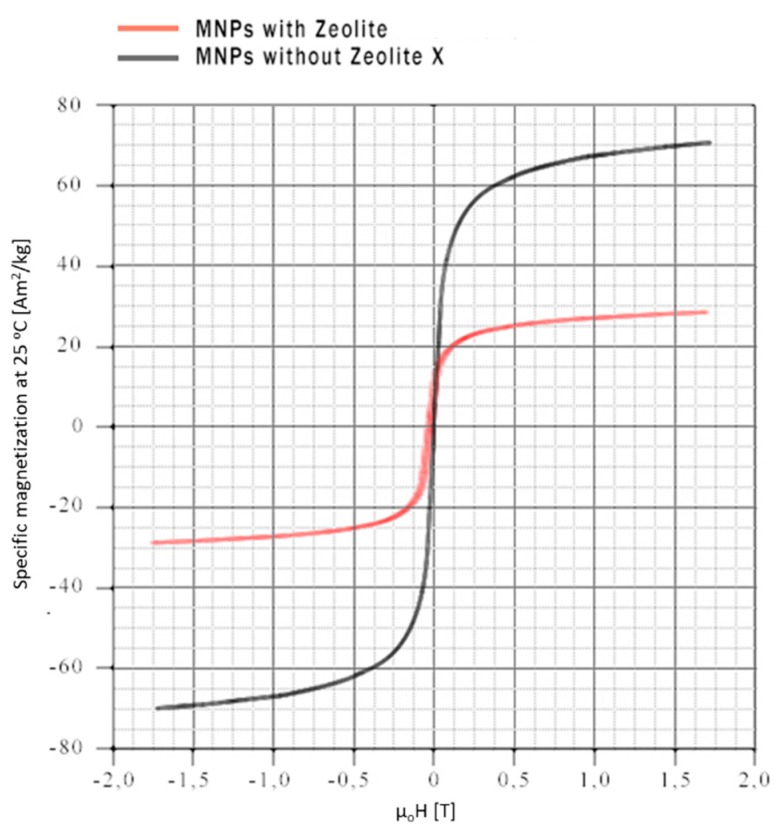
Hysteresis loops for the magnetite (MNPs) and the composite particles obtained by VSM.

**Figure 8 ijms-21-02707-f008:**
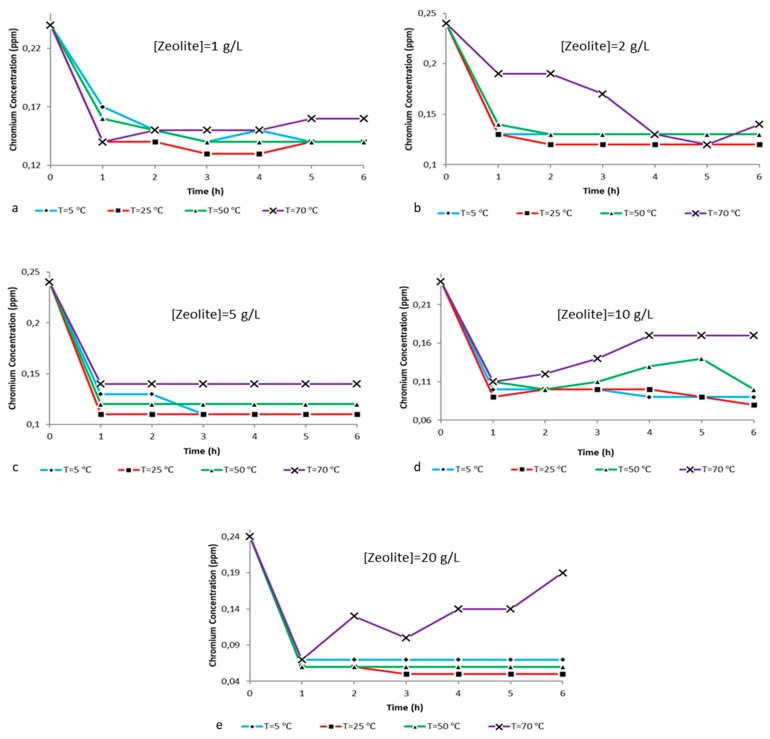
Chromium ion concentration vs time at different temperatures (5 °C, 25 °C, 50 °C and 70 °C) and pH 5.5 for the ratios (**a**) 2:1 g/L, (**b**) 2:2 g/L, (**c**) 2:5 g/L, (**d**) 2:10 g/L and (**e**) 2:20 g/L potassium dichromate:zeolite X.

**Figure 9 ijms-21-02707-f009:**
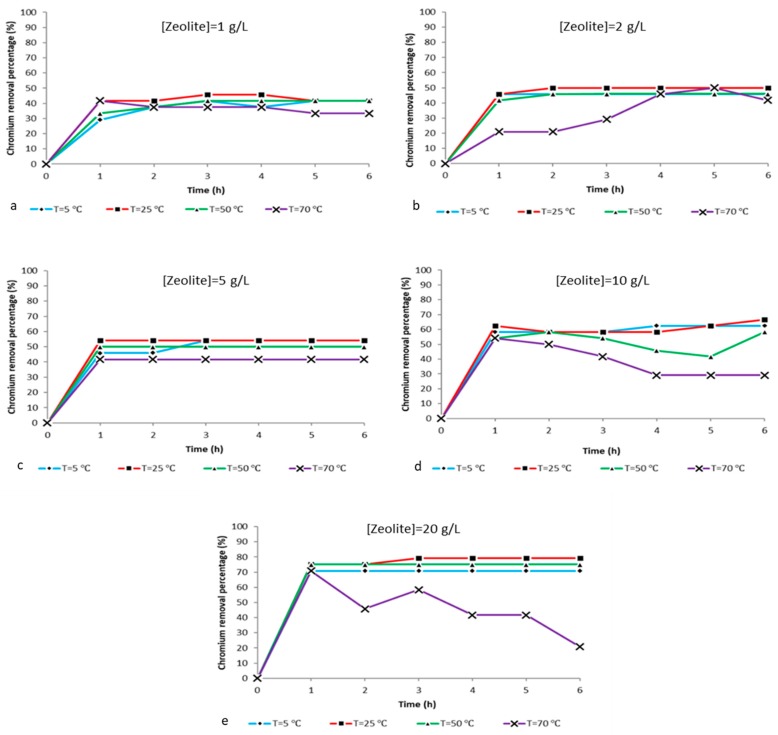
Chromium removal percentage vs time at different temperatures (5 °C, 25 °C, 50 °C and 70 °C) and pH 5.5 for the ratios (**a**) 2:1 g/L, (**b**) 2:2 g/L, (**c**) 2:5 g/L, (**d**) 2:10 g/L and (**e**) 2:20 g/L potassium dichromate:zeolite X.

**Figure 10 ijms-21-02707-f010:**
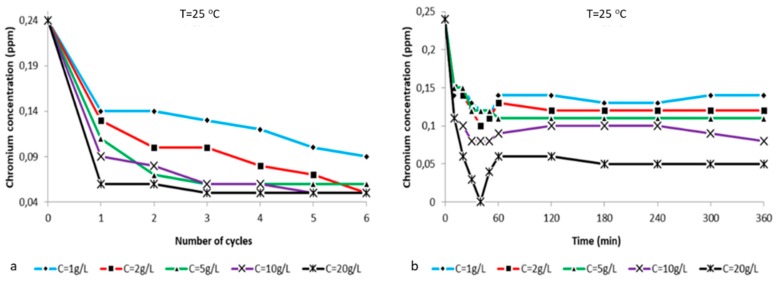
(**a**) Chromium ion concentration vs number of treatment cycles with zeolite X at T = 25 °C and pH = 5.5 for the ratios (i) 2:1 g/L, (ii) 2:2 g/L, (iii) 2:5 g/L, (iv) 2:10 g/L and (v) 2:20 g/L potassium dichromate:zeolite X. (**b**) Chromium ion concentration vs time (10 min for the first treatment hour and then every 1 h) at T = 25 °C and pH = 5.5 for the ratios (a) 2:1 g/L, (b) 2:2 g/L and (c) 2:5 g/L, (d) 2:10 g/L and (e) 2:20 g/L potassium dichromate:zeolite X.

**Figure 11 ijms-21-02707-f011:**
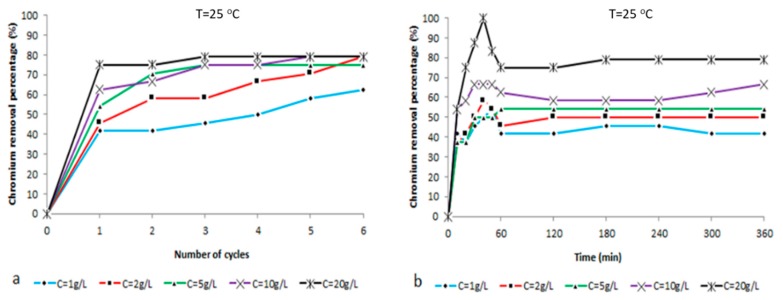
(**a**) Chromium removal percentage vs number of treatment cycles with zeolite X at T = 25 °C and pH = 5.5 for the ratios: (i) 2:1 g/L, (ii) 2:2 g/L, (iii) 2:5 g/L, (iv) 2:10 g/L and (v) 2:20 g/L potassium dichromate:zeolite X. (**b**) Chromium removal percentage vs time (10 min for the first treatment hour and then every 1 h) at T = 25 ° C and pH = 5.5 for the ratios: (i) 2:1 g/L, (ii) 2:2 g/L, (iii) 2:5 g/L, (iv) 2:10 g/L and (v) 2:20 g/L potassium dichromate:zeolite X.

**Table 1 ijms-21-02707-t001:** Chromium removal percentage vs number of treatment cycles with zeolite X at T = 25 °C and pH = 5.5 for the zeolite X concentrations: (i) 1 g/L, (ii) 2 g/L, (iii) 5 g/L, (iv) 10 g/L and (v) 20 g/L.

T = 25 °C	% [Cr^6+^] Reduction				
Time (Hours)	C = 1 g/L	C = 2 g/L	C = 5 g/L	C = 10 g/L	C = 20 g/L
**0**	0	0	0	0	0
**1**	42	46	54	63	75
**2**	42	58	71	67	75
**3**	46	58	75	75	79
**4**	50	67	75	75	79
**5**	58	71	75	79	79
**6**	63	79	75	79	79

**Table 2 ijms-21-02707-t002:** Chromium removal percentage vs time (10 min for the first treatment hour) at T = 25 °C and pH = 5.5 for the ratios: (i) 2:1 g/L, (ii) 2:2 g/L, (iii) 2:5 g/L, (iv) 2:10 g/L and (v) 2:20 g/L potassium dichromate:zeolite X.

T = 25 °C	% [Cr^6+^] Reduction				
Time (min)	C = 1 g/L	C = 2 g/L	C = 5 g/L	C = 10 g/L	C = 20 g/L
**0**	0	0	0	0	0
**10**	42	38	38	54	54
**20**	38	42	38	58	75
**30**	46	50	50	67	88
**40**	50	58	50	67	100
**50**	54	54	50	67	83
**60**	42	46	54	63	75
**120**	42	50	54	58	75
**180**	46	50	54	58	79
**240**	46	50	54	58	79
**300**	42	50	54	63	79
**360**	42	50	54	67	79

**Table 3 ijms-21-02707-t003:** The influence factors for the hexavalent chromium ion removal experiments.

Parameter	Experimental Change
**Time**	1–6 h
**Temperature**	5 °C, 25 °C, 50 °C, 70 °C
**Zeolite-X concentration**	1 g/L, 2 g/L, 5 g/L, 10 g/L, 20 g/L

**Table 4 ijms-21-02707-t004:** The experimental protocol for measurements made for the treated solutions with zeolite-X based on the influence factors.

Zeolite-X Concentration (g/L)	Temperature (°C)	Time (h)
**1**	5	1–6
	25	1–6
	50	1–6
	70	1–6
**2**	5	1–6
	25	1–6
	50	1–6
	70	1–6
**5**	5	1–6
	25	1–6
	50	1–6
	70	1–6
**10**	5	1–6
	25	1–6
	50	1–6
	70	1–6
**20**	5	1–6
	25	1–6
	50	1–6
	70	1–6

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
