# Peer review of "A Study on Magnetic Removal of Hexavalent Chromium from Aqueous Solutions Using Magnetite/Zeolite-X Composite Particles as Adsorbing Material"

_ijms, 2020, doi:10.3390/ijms21082707_

Round 1
Reviewer 1 Report
In the manuscript of “A study on magnetic removal of hexavalent chromium from aqueous solutions using magnetite/zeolite-X composite nanoparticles as adsorbing material”, authors describe a new type of composite material made by incorporation of magnetite in zeolite-X, and examine its metal-absorbing property as functions of concentration, medium temperature and temporal duration. Here are some comments on this work,
1.English of this manuscript needs to be over-hauled for clear readability of its content.
2.The general atomic structures of zeolite and magnetite are described Figs. 1 and 2, but the approximated structure for the composite is not well described, and how does the composite structure influence the metal-absorbing materials.
- The title of this manuscript emphasizes the removal of metallic ions by magnetics, however, does not reflect on any experimental study. Although magnetization is performed, how does it play a role in removing metallic ions, please explain. In a reference cited by authors, the magnetic property of iron oxide nanoparticles become paramagnetic when feature size is below 50 nm, please explain its consequence in the present study and its relevance to size calculation (equation 5) presented in this manuscript.
4. The FTIR spectra presented here is also not very informative, please be more specific.
5. Also, the description of discussion is a bit chaotic, please re-write it and make it more organized in terms of its context of experiment.
Author Response
In the manuscript of “A study on magnetic removal of hexavalent chromium from aqueous solutions using magnetite/zeolite-X composite nanoparticles as adsorbing material”, authors describe a new type of composite material made by incorporation of magnetite in zeolite-X, and examine its metal-absorbing property as functions of concentration, medium temperature and temporal duration. Here are some comments on this work,
Comment 1. English of this manuscript needs to be over-hauled for clear readability of its content.
Response 1. We extensively revised the manuscript; presentation is now clearer and languages appropriated
Comment 2. The general atomic structures of zeolite and magnetite are described Figs. 1 and 2, but the approximated structure for the composite is not well described, and how does the composite structure influence the metal-absorbing materials.
Response 2. Thank you for this comment. Please find the correction on lines 108-111.
Comment 3. The title of this manuscript emphasizes the removal of metallic ions by magnetics, however, does not reflect on any experimental study. Although magnetization is performed, how does it play a role in removing metallic ions, please explain.
Response 3. Thank you for this comment. Please find the correction on lines 171-176.
Comment 4. In a reference cited by authors, the magnetic property of iron oxide nanoparticles become paramagnetic when feature size is below 50 nm, please explain its consequence in the present study and its relevance to size calculation (equation 5) presented in this manuscript.
Response 4. Thank you for this comment. Please find the correction on lines 300-308.
Comment 5. The FTIR spectra presented here is also not very informative, please be more specific.
Response 5. Thank you for this comment. We revised the spectra in order to be more informative. The new spectra consist of 1 magnetite spectrum, 1 zeolite spectrum and 1 composite material spectrum. The peaks of interest are noted on the spectra. The text explains what each peak represents.
Comment 6. Also, the description of discussion is a bit chaotic, please re-write it and make it more organized in terms of its context of experiment.
Response 6. We changed the entire body of the text. We unified the results and discussion section and the text was revised and re-written so to be more understandable

Reviewer 2 Report
- Materials and methods part should be placed before results and discussion
- Try to obtain SEM images of better quality since they are not clear
- propose a mechanism for the removal process
- Check the English of the manuscript and correct the grammatical mistakes
- In figure 7, you mention that these experiments were done at different temperatures but you did not identify them
- Study the adsorption kinetics and isotherms
- Compare the obtained results in this study with those elsewhere in literature
- Study the possibility of adsorbent regeneration and reuse
Author Response
Comment 1. Materials and methods part should be placed before results and discussion
Response 1. Thank you for your comment. It is done.
Comment 2. Try to obtain SEM images of better quality since they are not clear
Response 2. We revised the SEM images. We added some more representative ones with better analysis at different focus.
Comment 3. propose a mechanism for the removal process
Response 3. Thank you for your comment. The removal process happens magnetically. We still investigate the appropriate set up for on line removal. Please check lines 171-176
Comment 4. Check the English of the manuscript and correct the grammatical mistakes
Response 4. We revised the language.
Comment 5. In figure 7, you mention that these experiments were done at different temperatures but you did not identify them
Response 5. Thank you for your comment. The temperatures are visible on the charts of figures 8 and 9 (i.e. T=25oC) and we supplemented them on the figure captions as well and on the charts of figures 10 and 11.
Comment 6. Study the adsorption kinetics and isotherms
Response 6. This reviewer raised a crucial point, we know that adsorption Kinect and isotherms are important parameters to be studied, we are already working towards this direction and these issues will be published separately elsewhere. Moreover, due the limitations we are facing with the pandemic it is hard for us to perfume new experiments. We hope the reviewer will understand.
Comment 7. Compare the obtained results in this study with those elsewhere in literature
Response 7. We added new text discussing the comparison, please check the text on lines 420-428.
Comment 8. Study the possibility of adsorbent regeneration and reuse
Response 8. We discussed this point at the end of the manuscript, please check the text on lines 445-451.

Round 2
Reviewer 1 Report
The authors have provided sufficient explanation and illustration after the revision, and I believe, it is now applicable for publication.
Reviewer 2 Report
The authors have addressed all the comments so the manuscript can be published in its present form